# Could Necroleachate Be the Cemetery’s Sewage? A Panorama from Brazilian Legislation

**DOI:** 10.3390/ijerph20196898

**Published:** 2023-10-08

**Authors:** Ana Paula Chein Bueno de Azevedo, Telma Abdalla de Oliveira Cardoso, Simone Cynamon Cohen

**Affiliations:** 1National School of Public Health, Oswaldo Cruz Foundation, Rio de Janeiro 21041-210, Brazil; abdalla.telma@gmail.com; 2Sanitation and Environmental Health Department, National School of Public Health, Oswaldo Cruz Foundation, Rio de Janeiro 21041-210, Brazil; simoneccohen@gmail.com

**Keywords:** cemetery, waste management, legislation, public health, Brazil

## Abstract

Cemeteries can be compared to landfills, as the leachate produced in these areas, also known as necroleachate, can be environmentally transported, polluting groundwater, surface water, and soil. In Brazil, to ensure no negative environmental impacts and public health risks, cemetery management is the responsibility of states and municipalities. In this context, this article aims to discuss Brazilian sanitary–environmental legislation concerning cemetery waste management. Only half of all Brazilian states have established sanitary–environmental cemetery legislation, and only 19 municipalities have specific laws. These laws, however, are broad and contain many gaps. Necroleachate care and control require both sanitary and environmental assessments to avoid environmental vulnerability and contamination risks for populations inhabiting surrounding areas. In this regard, new water analysis parameters in environmentally vulnerable areas should be established to control the population’s drinking water quality, such as the detection of *C. perfringens*. Furthermore, the construction of vertical cemeteries instead of horizontal ones and the adoption of cremation procedures should also be considered. This assessment comprises a novel research framework, as no studies on the impact of Brazilian laws on environmental necroleachate contamination are available to date.

## 1. Introduction

Mortuary practices date back to the Paleolithic archeology of the Ice Age and are considered a particularity of the human race, demonstrating how we deal with the dead and their corpses. The practice of burying the dead can be performed in two ways, namely inhumation and tombstoning. The former consists of burying bodies in open graves 1.5 m deep and covering the surfaces with earth and stone. The latter consists of burying corpses in masonry or concrete drawers, with a maximum depth of 5 m, which will later receive the coffins and, finally, be sealed [1].

Most historical accounts indicate changes in burial practices, especially regarding urban space land use and environmental sanitation, as body decomposition processes result in potential contaminants.

The scientific community’s concerns regarding the public health and environmental impacts of cemeteries from the 20th century led the United Nations to release an introductory summary on the subject. This summary explained how cemetery locations used to be chosen without proper planning and methodology, also without analyzing potential environmental or local community risks. It also addresses cemetery leachate’s impacts, a waste resulting from body degradation, on soil and groundwater and finalizes the document with suggestions in this regard [2]. In their report, Dent and Knight [3] compare cemeteries to a special type of landfill, where organic matter is filled in with soil. They, therefore, discuss the possibility of cemetery leachate being compared to landfill leachate, as it increases the concentrations of both organic and inorganic matter and the potential presence of pathogens, making adequate destinations paramount.

Cemetery leachate, also known as necroleachate, can be transported by rainwater infiltrating graves or by direct contact of the buried bodies with groundwater. This waste is made up of various components that, depending on internal and external factors, will facilitate or delay its contamination potential. 

The soil infiltration capacity depends on various intrinsic and extrinsic soil characteristics, such as the soil composition, hydraulic load, ion-exchange capacity, number of burials, and water table depth, among others [4]. It is, therefore, important to establish environmental laws concerning cemeteries to minimize their pollution potential.

Cemetery establishment and operation guidelines and laws are not universal, and each country has established its own environmental protection laws. In many European countries, for example, municipalities are responsible for establishing cemetery funeral practices and standards. This is the case in Denmark, where the municipality is the main manager, ensuring public funding for developments [5], while cemeteries are mainly managed by private companies in the United States (Table 1) [6]. 

It is, however, important to consider that most cemeteries are old and, therefore, non-compliant in terms of technical and environmental studies, both worldwide and in Brazil [11].

In Brazil, the disposal and allocation of corpses have been managed by the town halls of each municipality since the Proclamation of the Republic, respecting religious groups and local particularities. The Brazilian 1988 Federal Constitution, Art. 30, establishes that municipalities are responsible for organizing public services of local interest, promoting territorial planning, and controlling urban land use and occupation. Article 225 establishes the guarantee of all to a balanced environment essential to quality of life, ensuring laws for the installation of works or activities that may cause environmental degradation [12]. 

Given this scenario, where most existing cemeteries in Brazil, a continental-sized country, are old buildings, a lack of studies aiming at understanding the impacts of the broad and outdated legislation on the relationship between these constructions and environmental contamination by necroleachate is clearly noted. Thus, this article aims to discuss Brazilian sanitary–environmental legislation concerning cemetery waste management. Understanding the current laws and identifying successes and errors and what requires improvement are paramount for public and environmental health science professionals to create subsidies to improve cemetery management and minimize necroleachate contamination, guaranteeing better quality of life for surrounding communities and a healthier environment.

## 2. Material and Methods

This study applied a qualitative and exploratory approach. From this perspective, a documentary review of Brazilian legislation was used to develop the research.

Legal instruments were selected using databases available at the Brazilian federal government’s legislation portal [13] and state law and municipal law platforms [14,15]. These bases allow for searches of the entirety of Brazilian legislation. The keyword “cemetery” was used to locate established laws on the proposed topic.

The adopted inclusion criteria comprised all types of federal, state, and municipal laws and administrative acts enacted between 1970 and 2022 with the topic of cemeteries as the main scope and available in full. The beginning of the research was set at 1960, due to the founding of Brasilia, Brazil’s capital, expanding the research and the number of instruments to be analyzed. Exclusion criteria encompassed bills of law, as they are not currently being processed or have not yet been approved; federal, state, and municipal laws and administrative acts referring to animal cemeteries and without any specificity to the sanitary–environmental cemetery issue; duplicate legal instruments; and documents not available in full.

The searches were conducted in July 2023. Following the searches, all legal instruments were read in full, selected based on the aforementioned inclusion and exclusion criteria, and analyzed.

## 3. Results and Discussion

### 3.1. Environmental Legislation on Cemeteries 

Prior to the institution of specific environmental legislation for cemeteries established by the Ministry of the Environment’s National Environment Council (CONAMA), two legal instruments indicated the need for the licensing of establishments that use natural resources and could potentially pollute the environment, as follows. 

Federal Law No. 6.938 of 31 August 1981, which established the National Environmental Policy aimed at environmental protection. This law states that the construction, installation, expansion, and operation of establishments and activities that use environmental resources capable of causing environmental degradation will depend on environmental licensing [16]. This law, however, did not specify cemeteries as establishments that could potentially pollute or cause environmental degradation.

With the promulgation of the Federal Constitution, in 1988, the organization of public services of local interest, for the planning and control of the use and occupation of urban land, became the responsibility of Brazilian municipalities.

Thus, until 2003, no federal specific legislation or technical standards regulating the implementation and operation of cemeteries in environmental and sanitary terms were available. Therefore, the CONAMA legislation presented cemeteries as polluting sources and began to demand criteria for their construction and operations, aiming at environmental soil and groundwater protection [17] and minimizing contamination from cemetery facilities (Table 2).

In 2006, CONAMA Resolution 368 [19] amended Resolution 335 [18], due to the impossibility of building new cemeteries (or expanding existing ones) in certain municipalities in the states of São Paulo, Rio de Janeiro, Minas Gerais, and Bahia, installed in areas protected by water sources and on small plots of land, causing overcrowding. 

After two years, CONAMA Resolution 368 was updated by Resolution 402 of 2008, which set a new deadline for state and municipal environmental agencies to establish criteria for adapting cemeteries set up before the publication of Resolutions 335 and 368 to the new standards [20]. However, most Brazilian states and municipalities do not have adequate cemetery management tools, applying the CONAMA resolutions as the main environmental control instruments.

Table 3 depicts legal state frameworks identified herein encompassing some cemetery guidelines.

Of the 26 Brazilian states, half have established some type of approach to the health and environmental issue of cemeteries, while the rest have no reference to this issue in any legal instrument. In addition, only the law established by the state of Rio de Janeiro concerns cemeteries, and all others deal with the environment, health, and solid waste.

The states of Rio de Janeiro and Pernambuco exhibit the oldest legislation. The Pernambuco Decree-Law established general rules on health promotion, protection, and recovery, and it stipulates that all cemeteries should be under the supervision of the state health authority [29]. The Rio de Janeiro Decree is a specific instrument for cemetery establishment and operations. This legislation indicates that graves must be controlled to prevent the release of gases or odors so as not to contaminate the air, the underground water table, rivers, ditches, canals, and public roads. However, it does not describe what measures must be taken. It also determines that final burial constructions should be previously approved by the state authority, to be established in “public hygiene conditions” [30].

The Federal District (FD) also has specific legislation for cemetery construction and operations. A 1999 law applied by this state established location guidelines, stipulating that cemeteries should be built on high ground facing the water supply to cisterns, isolated from public places, and that the water table should be at least 2 m deep. It also stipulates that ornamental vases must be prepared so that they do not become water repositories that could serve as mosquito breeding sites [23].

In 1983 and 1986, the state of Santa Catarina enacted a law on general health regulations, which addressed specific criteria. This law stipulates that a cemetery can only operate following health authority approval, complying with this regulation’s rules regarding the construction project, installation and location, soil topography and nature, general hygiene and sanitation conditions, access roads, and urban planning. It also establishes building requirements, such as constructions in elevated areas, away from water that could feed wells and other supply sources, distant from public places and other open areas, and with a water table distance of at least 2 m in depth. Finally, it stipulates that construction projects must be accompanied by specialized studies that prove soil and water table suitability [32,33].

Five Brazilian states (Acre, Maranhão, Mato Grosso, Rondônia, and Tocantins) exhibit a general article in their legal instruments on environmental policy or groundwater administration and conservation. This law discusses that cemetery installations on the banks of springs can compromise minimum water quality standards. Furthermore, it requires the approval of health and environmental agencies concerning cemetery, crematoria, and morgue locations, constructions, installations, and operations [21,25,27,31,34]. The law in the state of Rondônia was slightly more specific regarding cemetery locations in spring basins, prohibiting their construction, installation, and operation within a 40 m radius of the banks of springs, waterholes, and water basins [31]. The state of Mato Grosso, in a legal instrument on groundwater conservation, addressed corpse transfer aspects in the state’s territory and dictated the need for health authorization [27].

In a law on groundwater deposit conservation and environmental protection, the state of Goiás established criteria for the prevention and monitoring of groundwater resource pollution in cemeteries. This law determines the implementation of water quality monitoring wells, establishing their constituent elements, and determining requirements for cemetery construction and operations, such as environmental licensing, describing the local geology, and determining groundwater flow directions. It also defines the need for a report to be submitted to the state’s environmental monitoring agency containing information on the geological profile of the monitoring wells, as well as their casing, filters and pre-filters, buffers, and operating systems, and a description of the monitoring and sanitary protection system. It also points out that, if necessary, the environmental monitoring agency may require the use of geophysical methods applicable to the prevention of underground water resource pollution for the storage or disposal of liquid and solid waste [24].

In a law establishing the state’s solid waste policy, Bahia classifies cemetery waste as solid waste and categorizes it separately from healthcare waste [22].

Mato Grosso do Sul and Minas Gerais have both enacted laws that deal with the burial, cremation, embalming, exhumation, transportation, and display of corpses—in 1992 and 2012, respectively [26,28]. 

The Mato Grosso do Sul law establishes the state’s health code. It stipulates that the deposit and handling of corpses for any purpose, including necropsies, embalming, or any other procedures for corpse preservation, can only be conducted in health department-authorized establishments. Corpse cremations and burials can only take place in health authority-licensed cemeteries that exercise health surveillance over funeral facilities and services. These authorities can order the execution of works for cemetery sanitary improvements as well as their temporary or definitive banning. The law also defines that the transfer and deposit of human remains or their ashes requires a health permit [28].

The Minas Gerais law deals with sanitary and environmental burial conditions. It determines the use of techniques and practices applied in cemeteries that allow for gas exchange and prevent their passage to areas where people circulate and the leakage of liquids from colliquation, to minimize environmental damage. The law also determines a minimum distance for graves of 1.5 m above the highest water table level and stipulates that funeral blankets or urns must be made of biodegradable materials [26].

As CONAMA Resolution No. 368 of 2006 established the obligation of state and municipal bodies to license and inspect the construction and implementation of new cemeteries, another database search was performed to identify the legal municipal framework for cemeteries [19].

According to the 2010 Census, Brazil has 5565 municipalities, of which only 38 have more than 500,000 inhabitants [35]. We could only retrieve specific cemetery laws from 66 municipalities, or 1.19% of the total, due to difficulties in accessing the data and the absence of specific legislation. Of these 66 municipalities, 47 do not mention specific articles on health and environmental issues in their laws, leaving legal guidelines to the CONAMA resolutions alone. The other 19 municipalities, in addition to following the CONAMA resolutions, present legal instruments from municipal environmental departments. Their specific articles are described in Table 4.

All identified municipal laws point to issues concerning potential water and groundwater contamination by decomposing bodies. The Sorocaba (SP), Araraquara (PR), Caratinga (MG), Dom Viçoso (MG), Planalto da Serra (MT), and Rio Branco (AC) laws also detail the need to adopt necessary and sufficient measures to avoid such contamination, such as technical studies, the characterization of future cemetery areas (locations, topographic surveys, aquifer level studies, mechanical drilling for soil characterization), planning for the implementation and operation of the project, environmental licensing, groundwater monitoring, and a neighborhood impact study [36,41,45,46,54,55].

To prevent these problems, the laws of the municipalities of Sorocaba, Diadema, Praia Grande, and Americana (SP); Santa Terezinha de Itaipu and Quedas do Iguaçu (PR); and Planalto da Serra (MT) and Rio Branco (AC) stipulate that cemeteries cannot be built above reservoirs, water supply systems, or spring basins and that they must be located at least 2 m away from the deepest point used for graves [36,37,38,39,47,48,54,55]. The Praia Grande (SP) law also points out that if the above specifications cannot be fulfilled, the groundwater level should be lowered by drainage processes. If drainage is not possible, the thickness of the chamber should be increased by raising the land surface through earthworks [38]. The laws of Sorocaba (SP), Caratinga (MG), and Planalto da Serra (MT) also require cemetery grounds to be sufficiently elevated to prevent the possibility of grave flooding [36,45,54].

The laws of the municipalities of Montes Claros (MG), Cuiabá, and Colíder (MT) point out that so-called “shallow graves” can be used at a minimum depth of 1.55 m in flat lands in cemeteries [44,52,53]. In addition, the Rio de Janeiro, Teresópolis (RJ), and Cuiabá, Colíder (MT) laws point out that burials in drawers, consoles, or shelves, below or above ground level, in traditional cemeteries will only be allowed when performed under “satisfactory public hygiene conditions” [42,43,52,53].

### 3.2. Cemetery-Generated Waste 

Cemeteries can generate waste from grave construction and maintenance, dry and green waste from floral arrangements, wood waste from skiffs, and body decomposition waste from bones and others and from the exhumation process. Due to its characteristics, cemetery waste management is important, as inadequate management can result in environmental and public health impacts, in view of the possibility of spreading pathogenic biological agents and affecting human health and quality of life, especially for those who work in cemeteries and live in the surrounding areas [56,57,58,59,60,61,62].

Federal Law No. 12.305 was enacted in 2010, establishing the National Solid Waste Policy and laying down guidelines for solid waste management in Brazil, including the dangers and responsibilities of waste generators [63]. According to this policy, solid waste is defined as discarded solid or semi-solid material resulting from human activities in which final disposal must be managed. These include the gases contained in containers and liquids whose particular characteristics make it impracticable to discharge them into the public sewage system or directly into water sources [63]. In addition, this policy classifies solid waste according to its hazard and origin. Regarding the former, solid waste can be categorized as hazardous, whose flammability, corrosivity, reactivity, toxicity, pathogenicity, carcinogenicity, teratogenicity, and mutagenicity characteristics lead to significant public health or environmental risks, and non-hazardous, originating from domestic activities in urban residences (household waste) [64]. 

According to waste origin, Federal Law No. 12.305 establishes that household waste and urban cleaning waste, originating from sweeping and cleaning public places, roads, and other urban cleaning services, should be classified as solid urban waste [64]. However, it does not specify which waste is hazardous or non-hazardous, nor does it include cemetery waste in the solid urban waste category.

Brazilian Standard (NBR) No 10.004, established by the Brazilian Association of Technical Standards (ABNT), dealt with this issue in 2004 and defined solid waste as any solid or semi-solid waste resulting from “industrial, domestic, hospital, commercial, agricultural, service and sweeping activities”. It also included sludge originating from water treatment systems and generated in pollution control equipment and installations, as well as certain liquids whose particularities make it unfeasible to discharge them into the public sewage system or water sources. This law also classifies waste, based on hazardousness, according to physical, chemical, or infectious properties, which can pose public health risks, resulting in death and/or disease, and environmental risks when managed incorrectly. In this sense, two types of waste are noted, Class I waste, covered by Law No. 12.305 [63], and Class II waste, resulting from several activities that do not present risks [64].

In this regard, solid cemetery waste can be classified as Class I or Class II waste. Class I waste, depending on the type, may contain biological agents or toxins capable of producing diseases in humans and animals, while Class II waste can be assimilated into construction or municipal solid waste, as it contains waste resulting from the construction, renovation, repair, and demolition of civil works carried out in cemeteries and burial grounds, as well as from the preparation and excavation of grave plots and from recyclable materials, such as plastic from artificial flowers, vases, and fabrics and those from sweeping, tree pruning, and leaves and natural flowers generated in urban cleaning activities [65,66]. 

Moreover, in 2004, the National Health Surveillance Agency (ANVISA) established the Collegiate Board Resolution (RDC) 306 health legislation [67] concerning healthcare waste, which was updated in 2018 by RDC 222 [68]. It should be noted that the 2005 publication of CONAMA Resolution 358 used the same technical criteria but addressed environmental aspects [69]. According to these laws, cemetery waste, solid and liquid, is classified as healthcare waste. 

Services that generate healthcare waste encompass all activities related to human or animal health care, including “home care services; (...) morgues, funeral homes, and services where embalming activities are conducted (thanatopraxia and somatoconservation) and forensic medicine services; (...)” [67,68,69]. 

These wastes can be classified into the following five groups [68].

Group A: may contain possible biological agent contamination, such as microorganism cultures and stocks, vaccines, animal waste, anatomical human parts, arterial line kits, air filters and gases aspirated from contaminated areas, laboratory samples, waste from surgeries and health care, sharp or scarifying materials, and other materials resulting from the health care of humans or animals. 

Group B: waste containing chemical products dangerous to public health or to the environment, depending on their flammability, corrosiveness, reactivity, toxicity, carcinogenicity, teratogenicity, and mutagenicity characteristics and amounts.

Group C: waste that contains radionuclides at higher levels than those determined by Brazilian standards.

Group D: waste that does not present biological, chemical, or radiological risks to human health or the environment and can be assimilated into household waste.

Group E: sharp or scarifying waste (needles, glass ampoules, etc.). 

Based on this classification, the main waste found in cemeteries is described in Table 5.

Cemetery waste can fall into either Group D or E, which pose lower environmental risks, or into Groups A and B, where it poses higher risks due to the presence of biological and chemical agents that can contaminate the environment if not treated correctly, whether liquid or solid [70].

As indicated in Table 2, only one state’s legislation has addressed the issue of cemetery waste, namely Law No. 12.932 of 2014, from the state of Bahia, where it subdivides cemetery waste into human (derived from cadaveric exhumation) and non-human (derived from periodic cemetery cleaning and maintenance) waste [22]. 

According to Table 3, Planalto da Serra (MT) is the only municipality with a municipal law indicating that solid non-human waste resulting from body exhumation must be disposed of in an environmentally appropriate and sanitary manner [54].

### 3.3. Necroleachate

Necroleachate is a viscous liquid, of varying colors, percolated from the decomposition of corpses. This waste has a strong unpleasant smell and is usually formed between weeks and months after death. 

A decomposing human body weighing between 70 kg and 80 kg can release around 30 L of necroleachate. This percolate comprises 60% water and the remainder consists of mineral salts and degradable organic substances, as well as potential pathogens (Table 6) [3,8,70,71]. 

Necroleachate’s composition can promote the survival and proliferation of microorganisms arising from decomposition, and it may contain mineral loads and released by-products, as well as biogenic amines like putrescine (C_4_H_12_N_2_: 1.4-butanediamine) and cadaverine (C_5_H_14_N_2_: 1.5-pentanediamine), both of which exhibit high pathogenic loads. The degradation of these compounds generates ammonium (NH_4_), nitrite (NO_2_), nitrate (NO_3_), carbon monoxide (CO), carbon dioxide (CO_2_), and methane (CH_4_). At high concentrations, these substances are highly pollutant and toxic to humans. In addition to decomposition, other components, such as radiological, chemotherapeutic, and embalming elements (e.g., arsenic, formaldehyde, and methanol), make-up (e.g., cosmetics, pigments, and compounds), as well as dental fillings and heart pacemakers [58,62,72,73,74,75,76,77,78], may also be present. 

Ueda et al. [79] collected samples from cemeteries in the city of São Paulo, in Southeastern Brazil, over a period of 3 years, isolating the main microorganisms present in corpses that can lead to soil contamination and become disease transmission vehicles associated with corpse decomposition (Figure 1). These include the causative agents of tetanus (*Clostridium tetani*), gas gangrene (*C. perfringens*), toxic food contamination (*E. coli*, *Streptococcus*), tuberculosis (*Mycobacterium tuberculosis*), paratyphoid fever (*Salmonella paratyphi*), bacterial dysentery (*Shigella dysenteriae*), and cholera (*V. cholerae*) [2,58,74]. 

Necroleachate thus alters the physicochemical and biological soil and groundwater characteristics in areas where no necroleachate waste treatment or collection facilities are available [75,77]. Because of this, cemeteries represent a source of sanitary and environmental liability [57,58,73,80,81]. 

In addition, cemeteries have been historically built in areas presenting inadequate geoecological, hydrological, and geotechnical conditions from a legal point of view, increasing the socio-environmental health vulnerability of surrounding areas [82,83,84]. 

It is also important to note the existence of by-products originating from the decomposition of coffins and their props, as well as the fabrics used to dress bodies and coffin beds. The employed fabrics are made from hard-to-degrade materials and are treated with chemical binders. The wood used for coffins is usually treated with preservatives such as polyvinyl chloride, creosote, or insecticides, as well as varnishes and sealants, which release harmful toxic substances during their degradation [62,75,75,85,86]. 

Several authors point out the presence of heavy metals, which can expose surrounding populations to heightened risks, as depicted in Table 7 [4,62,74,87,88,89,90,91]. 

For this reason, necroleachate is classified as a Class I hazardous waste and as a healthcare waste. The establishment of criteria and requirements for adequate cemetery construction and maintenance is, therefore, clearly paramount, as necroleachate cannot be discharged into public sewage systems or water sources without prior treatment [64].

Of the 13 Brazilian states that address the possibility of water table necroleachate contamination, 10 discuss aspects related to grave construction concerning water table location (AC, DF, GO, MA, MG, MT, RJ, RO, SC, and TO) [21,23,24,25,26,27,30,31,32,33,34]. The states of SC, DF, and MG point out the need for graves to be located at a minimum distance from the water table. DF and SC indicate 2 m [23,32,33], while MG determines 1.5 m [26]. GO is the only state that has addressed issues related to the groundwater resource monitoring pollution system [24]. 

Point sources, such as cemeteries, usually produce defined and concentrated contamination plumes, which makes them easier to identify [74]. This consists of the introduction of substances that cause harmful alterations regarding aquatic environment use, thus characterizing pollution [92]. 

As necroleachate is denser than water, it exhibits heightened mobility and dispersion, crossing aquifers up to impermeable layers and being partly carried in the direction of the underground flow, contaminating entire regions. 

The severity of groundwater contamination partly depends on the waste or leachate characteristics, i.e., volume, composition, the concentrations of their various constituents, contaminant release time rates, the size of the area from which the contaminants are derived, and leachate density, among others [4,93]. 

The delay and dispersal mechanisms concerning the migration of chemical and microbiological contaminants in soil cover and rock environments also depend on factors such as the local climate and geology, type of soil, hydraulic soil conductivity, cation exchange capacity, water table depth, water flow patterns, and the nature of the contaminating agents. Moreover, different chemical reactions, such as chemical precipitation, chemical, radioactive, and biological degradation, volatilization, biological consumption, and adsorption, can interfere with infiltration to deeper soil zones, preventing or degrading contaminants [4,93]. 

The water affected by necroleachate results in high oxygen consumption, due to the biological decomposition and chemical transformation of products containing nitrogen, phosphorus, and sulfur, among others. This leads to more mineral salts, increasing the electrical conductivity of affected waters [94]. Assessments concerning these parameters will guide possible groundwater necroleachate alteration or contamination evaluations, and the presence of previously mentioned microorganisms can comprise important necroleachate bioindicators [2,58,74,95,96]. 

Another important aspect in guaranteeing human health and environmental control concerns the requirements for necroleachate draining infrastructures. This, however, is rarely addressed in current legislation. Depending on the area, high rainfall rates and a lack of maintenance increase the spread of necroleachate waste [61], which may be avoided by employing adequate drainage conditions, thus reducing necroleachate runoff [97]. 

The only two municipal laws in Brazil that address this issue are implemented in São Francisco de Paula (RS) and Sorocaba (SP). The former stipulates that all graves or overlapping drawers must have an individual drainage system and describes specifications for the installation of PVC pipes or similar equipment 40 mm in diameter from the grave to a 40 × 40 × 40 cm length/height/width drainage box filled with gravel. Sorocaba (SP), on the other hand, requires the installation of drainage system pipes with a minimum diameter of 50 mm and a septic tank to receive the liquid waste originating from body decomposition and the washing water from the drainage system [36,49].

Some municipal laws indicate the type of material that should be used in the manufacture of urns and coffins to avoid soil and groundwater contamination when these items decompose. The municipalities of Sorocaba (SP), Teresópolis (RJ), Montes Claros (MG), São Borja (RS), Planalto da Serra (MT), and Rio Branco (AC) prohibit the use of metal coffins or wooden coffins covered internally or externally with metal, except in cases of embalming, exhumation, and autopsied corpses. Autopsied corpses, limbs, or viscera must be deposited in zinc or tinplate coffins. In addition, these municipalities determine that inhumed corpses must be wrapped in necroleachate-absorbing wrappings and funeral urns must be made of biodegradable materials and hermetically sealed. Decomposition-accelerating materials or purification filters must be present inside these urns, along with devices that prevent the effects of interior gas pressure [36,43,44,50,51,54,55].

The possibility of air contamination is noted through the combined action of corpse autolysis and putrefaction, leading to drastic physical and chemical changes. Macromolecules are degraded, resulting in the emission of hundreds of different volatile organic compounds (VOCs), such as dimethyl sulfide, dimethyl disulfide, toluene, hexane, and dimethyl trisulfide [98,99]. Studies have emphasized the influence of several factors on VOC release from corpses, including the environmental conditions (temperature, humidity, air currents), soil type, scavenging insects, and decomposition stages [100]. In addition to pollution, cadaveric odor also plays a key role in attracting scavenging insects and other vertebrate scavengers [101]. Houseflies (*Musca domestica*), as with many other necrophagous insects, are considered worldwide pests. Besides being irritants, they are also vectors that can transmit numerous human and animal diseases caused by many antibiotic-resistant zoonotic pathogens [102,103]. 

Odor and air quality impacts can reach several hundred meters downwind and result in inflammatory and immunological effects or irritation to the eyes, nose, or throat [104,105]. The release of bioaerosols, particulates, or VOCs and their potential harmful or health effects have not been widely assessed and may represent an area for further assessment [106].

The laws of the municipalities of Sorocaba, São Paulo (SP); Rio de Janeiro, Teresópolis (RJ); Dom Viçoso (MG); São Borja (RS); and Cuiabá and Colíder (MT) indicate that graves must be prepared to avoid the release of putrid gases or odors that could pollute or contaminate the air. The use of impermeable material that prevents the gaseous exchange of buried bodies is, however, prohibited [36,40,42,43,46,50,51,52,53].

## 4. Final Considerations

In Brazil, no control over the construction of cemeteries is noted, which is, in turn, aggravated by a lack of monitoring by different competent public authority spheres, who redirect the problem towards one another. 

The main negative environmental impacts of cemetery waste, including necroleachate, are related to non-compliance with current regulations, a lack of supervision, environmental management, and a multidisciplinary team of trained professionals. 

To aid in the controversial discussion on the subject, it is believed that the problem lies not only in the activity itself but also in the mistaken and careless selection of cemetery locations. Many of them are environmentally vulnerable, which entails soil, groundwater, and surrounding human population contamination risks. The literature on the subject is, however, limited. 

Groundwater contamination is a major global concern. In developing countries, groundwater is subject to contamination due to poor sanitation associated with high population growth rates. In rural areas of developed countries, groundwater contamination is attributed to fertilization, pesticide and herbicide use, and inadequate irrigation habits from the point of view of the amounts of groundwater pumping and water used for irrigation. In urban areas of developed countries, groundwater contamination is attributed to a variety of aspects (e.g., chemical pollutants and microorganisms) originating from urban development and domestic and industrial waste.

The multiplicity of substances that comprise necroleachate and the potential for contamination justify the importance of environmental licensing, where cemeteries must present technical studies that prove their viability in future construction areas. It is also relevant to recommend environmental analyses that consider aspects related to pedological and hydrogeological characteristics when choosing cemetery sites. These characteristics include issues related to the terrain topography, relief, slope, geology, type of soil and its hydraulic conductivity, hydrology, level, piezometric surface positioning, liquid water table recharge, and subsurface runoff flow direction and speed. These factors are essential in understanding pollutant dynamics in the unsaturated zone, as soil acts as a filter in retaining chemical, microbiological, and other substances resulting from the corpse decomposition process. Regional regulations that consider pedology and hydrogeology are required in Brazil, due to the vastness of the Brazilian territory. The country has established water quality legislation concerning microbiological standards for drinking water and bathing water quality, in the form of the Ministry of Health’s Ordinance No. 2.914 and CONAMA Resolutions 357 and 396 [107,108,109]. The presence of *C. perfringens* is not, however, considered in this legal framework, but it poses a potential risk to water users in areas near cemeteries. This demonstrates the need to include specific parameters for the analysis of water samples intended for human consumption in environmentally vulnerable areas such as cemeteries. 

Horizontal cemeteries are similar to landfills, often a polluting activity, but are still the preferred means of dealing with corpses in Brazilian municipalities. Although the establishment of a legal framework that considers construction, location, and regulation aspects is a very positive factor, many cemeteries are still categorized as being in unsuitable conditions because they were built prior to the establishment of this legal framework, although their activities have not been banned. The environmental risks that these cemeteries entail are extremely high, considering that the possibility of environmental impacts is high and so are their consequences.

Given this scenario, we suggest solutions to minimize the environmental impacts caused by the main cemetery waste, necroleachate, including the implementation of vertical cemeteries and crematoria, instead of horizontal or traditional cemeteries.

Vertical cemeteries prevent necroleachate from coming into direct contact with the ground through drainage, drying, and treatment systems, leaving approximately 50 g of solid matter at the end of the process, which is sent for final disposal as healthcare waste. This avoids groundwater pollution and does not compromise water quality. These structures also allow for the treatment of gases released by decomposing corpses through molecular dissociation, through gas exchange and filtering throughout the decomposition process. The implementation of vertical cemeteries should be, therefore, prioritized to attend to a high number of burials in dense urban areas.

Another architectural structure that offers solutions to the necroleachate challenge comprises crematoria. These structures reduce corpse volumes through a combustion process conducted at temperatures ranging from 482 to 797 °C [110]. Neckel et al. [57] demonstrated that a 70 kg corpse is reduced to 1 kg of ash and bone fragments. However, the design must incorporate suitable equipment to filter the exhaust from the combustion chamber and limit the release of particulate matter into the surrounding atmosphere during the combustion process.

These solutions, however, require periodic maintenance to keep all systems working properly, which is not necessary in a horizontal necropolis. As such, the vertical system presents higher maintenance costs than the horizontal one for public administrations.

This is a subject shrouded in dogmatic issues such as the cremation process, and, on the one hand, bodies buried in horizontal cemeteries can lead to soil and surface and underground water contamination, requiring both environmental and sanitary care and control; perhaps because it borders these two areas, the laws are not specific. These issues demonstrate the complexity and challenges of future solutions.

The results of this study demonstrate that Brazilian state legislation, as well as federal laws, exhibits many gaps. The criteria required to manage cemeteries are sometimes covered by environmental legislation, to control, prevent, and monitor water resource pollution, and, at other times, by health legislation. Furthermore, most laws are too broad. We hoped that the Brazilian states would present more detailed legal frameworks than the federal framework, as regional aspects that should be considered when designing cemetery projects, such as hydrogeological issues, are noted, although this was not the case.

## Figures and Tables

**Figure 1 ijerph-20-06898-f001:**
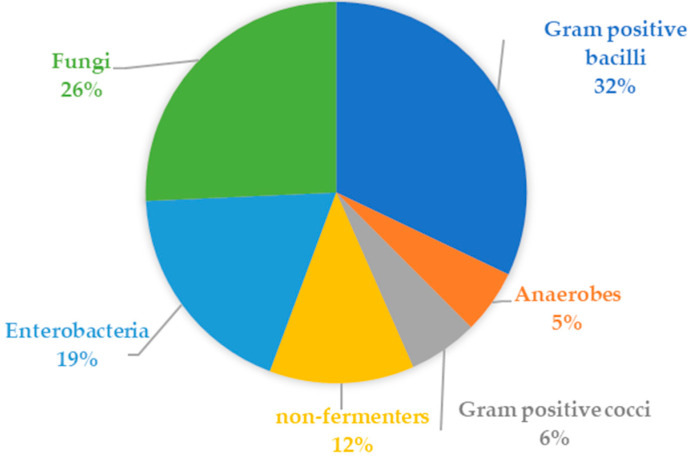
Microorganisms isolated from cemetery soil in São Paulo, Brazil. Adapted from Ueda et al. [80].

**Table 1 ijerph-20-06898-t001:** Special notes concerning cemetery establishment and operation guidelines worldwide.

Country	Notes
Nigeria [7]	Cemetery laws and operations follow Islamic religious rules. Thus, certain practices are required, such as not embalming corpses, or using cotton as wrapping to increase body decomposition. In addition, cemeteries should have preserved green areas, and graves should be built as far away from water tables as possible.
Vietnam [8]	According to Vietnam’s Construction, Management, and Employment of Cemeteries and Burial Land legislation, all constructions should be periodically maintained. In addition, they must have plans for land use, regional construction, urban construction, and rural residential zones. Moreover, when building graves, constructions must implement a plate system using high-density polyethylene (HDPE) alongside a leachate collection system to prevent leachate from reaching groundwater and deeper aquifers.
United Kingdom [9]	The UK’s Town and Country Planning Act 1990 covers the development of new cemeteries and the expansion of existing ones. In this sense, certain practices should be followed for groundwater protection, such as avoiding human burials in floodable lands, building grave bases at least 1 m above the highest water table level, and not building graves in unaltered or intact rock.
United States of America ^1^ [10]	USA cemetery operations should guarantee health and general welfare safety and protection. Thus, the construction of new cemeteries or the expansion of existing ones must consider whether the selected area contains drinking water, sewage disposal, refuse disposal, and highlands to ensure that the local groundwater will not be contaminated by cemetery activities.

^1^ These data are available only for Conrad Broward, Florida.

**Table 2 ijerph-20-06898-t002:** Environmental federal legislation specific to cemeteries in Brazil.

Legislation	Date	Note
CONAMA Resolution 335 [18]	2003	Provides for the environmental licensing of cemeteries and issues involving construction aspects and cemetery regulations, such as Horizontal and vertical cemeteries must be subjected to environmental licensing processes; Horizontal cemeteries must be established at a minimum distance of 1.5 m from the maximum local aquifer level and, when not possible, burials must be conducted above ground level; Techniques and practices must be adopted to allow for gas exchange, providing adequate conditions for body decomposition; Buried bodies may be wrapped in blankets and urns may be made of biodegradable materials, but they may not use plastics, paints, varnishes, heavy metals, or other impermeable materials; and Non-human solid waste resulting from body exhumation must be disposed of in an environmentally appropriate and sanitary manner.
CONAMA Resolution 368 [19]	2006	Deals with the environmental licensing of horizontal and vertical cemeteries, based on certain particularities of water source protection areas located in metropolitan areas. Mandates that cemeteries already established at the date of publication of this Resolution have up to 2 years to adapt to the resolution’s rules. For example, No cemeteries can be constructed in permanent preservation areas and in areas where the water table is less than 1.5 m from the bases of the cemetery graves; State and municipality bodies are obliged to license and inspect the construction and implementation of new cemeteries.

**Table 3 ijerph-20-06898-t003:** Brazilian state legislation.

State	Legislation	Year	Content
Acre	Law No. 1117 [21]	1994	The state’s environmental policy
Bahia	Law No. 12.932 [22]	2014	The state’s solid waste policy
Federal District	Law No. 2.424 [23]	1999	Construction, operation, use, administration, and supervision of cemeteries and the performance of funeral services
Goiás	Law No. 13.583 [24]	2000	Conservation and environmental protection of groundwater deposits in the state
Maranhão	Law No. 7.587 [25]	2001	Prohibits the installation of projects located on the banks of water sources
Minas Gerais	Law No. 20.017 [26]	2012	Sanitary and environmental conditions for burials in the state
Mato Grosso	Law No. 9.612 [27]	2011	Administration and conservation of groundwater in the state’s domain
Mato Grosso do Sul	Law No. 1.293 [28]	1992	The state’s sanitary code
Pernambuco	Decree-Law No. 268 [29]	1970	General rules on health promotion, protection, and recovery
Rio de Janeiro	Decree “E” No. 3.707 [30]	1970	Guidelines for cemetery establishment and operations
Rondônia	Law No. 1.878 [31]	2008	Installation of projects in water source basins
Santa Catarina	Law No. 6320 [32]	1983	General health standards
	Decree No. 30.570 [33]	1986	Regulates articles 48, 49, and 50 of Law No. 6.320
Tocantins	Law No. 261 [34]	1991	The state’s environmental policy

**Table 4 ijerph-20-06898-t004:** Brazilian municipal cemetery legislation.

Municipality	State	Legislation	Content
Sorocaba	São Paulo	Law No. 5.271 [36]	Cemetery operations
Diadema		Municipal Law No. 3.048 [37]	Cemetery practices
Praia Grande		Supplementary Law No. 606 [38]	Cemetery installation and operation guidelines
Americana		Law No. 5.750 [39]	Cemetery establishment, management, and use guidelines
São Paulo		Decree No. 59.196 [40]	Funeral, cemetery, and cremation guidelines
Araraquara		Supplementary Law No. 971 [41]	Cemetery and crematoria installation, organization, and operation guidelines
Rio de Janeiro	Rio de Janeiro	Decree No. 39.094 [42]	Cemetery and funeral regulations
Teresópolis		Municipal Law No. 4.079 [43]	Public and private cemetery establishment, administration, operation, and use guidelines
Montes Claros	Minas Gerais	Law No. 3.800 [44]	Cemetery construction, operation, use, administration, and supervision and funeral service guidelines
Caratinga		Law No. 3.626 [45]	Use of the municipal cemetery of Caratinga
Dom Viçoso		Supplementary Law No. 1.149 [46]	Facilities, standards, and procedures to be followed in cemeteries and mortuary chapels
Santa Terezinha de Itaipu	Paraná	Law No. 75 [47]	Cemetery services
Quedas do Iguaçu		Law No. 945 [48]	Cemetery and funeral services
São Francisco de Paula	Rio Grande do Sul	Law No. 2.757 [49]	Cemetery and funeral service activities, use, and provisions
São Borja		Law No. 5.810 [50]	Regulation of funeral, cemetery, and cremation services
		Decree No. 19.353 [51]	Regulation of Law No. 5.810
Cuiabá	Mato Grosso	Law No. 2.339 [52]	Creation, construction, and operation of public and private cemeteries
Colíder		Law No.167 [53]	Creation, construction, and operation of public and private cemeteries
Planalto da Serra		Law No. 558 [54]	Municipal and private–public cemetery regulations and regularization guidelines
Rio Branco	Acre	Law No. 1.809 [55]	Cemetery and funeral service rules

**Table 5 ijerph-20-06898-t005:** Cemetery waste categorized according to Brazilian laws.

Waste Classification	Waste
Group A	Waste from body exhumation and grave cleaning, both the product of dry settling and other materials (decaying wood, mortuary bags, blankets, quilts and coffin lining materials, bones, jewelry, leftover clothing, protective equipment such as gloves, etc.)
Group B	Paint and/or varnish cans and other flammable waste, fluorescent lamps, batteries
Group D	Natural flowers, wreaths, arrangements, ornamental objects, plastic vases, landscaping waste, sweeping and tree pruning, plastic cups, paper, packaging, plastic, candle wax, and construction debris (bricks, ceramic blocks, concrete in general, mortar, stone, wood, soil, rocks, glass, plastic, pipes, electrical wiring, etc.)
Group E	Glass vases, ceramic images, and metal objects

**Table 6 ijerph-20-06898-t006:** Necroleachate composition.

Necroleachate Composition [3,71]	
Substance	Mass (g)
Oxygen	43,000
Carbon	1600
Hydrogen	7000
Nitrogen	1800
Calcium	1100
Phosphorus	500
Sulfur	140
Potassium	140
Sodium	100
Chlorine	95
Magnesium	19
Iron	4.2
Cooper	0.07
Lead	0.12
Cadmium	0.05
Nickel	0.01
Uranium	0.00009
Total body mass	70,000

**Table 7 ijerph-20-06898-t007:** Heavy metals and risks.

Heavy Metals	Risks/Disabilities
Cu	Neurotoxicity
Zn	Liver disease
Fe	Gastrointestinal tract corrosion and liver cirrhosis
Pb	Central nervous system toxicity
Cr	Lung cancer in humans
Cd	Osteoporosis
Ni	Classified as a human carcinogen

## Data Availability

The databases consulted to write this article are available at the Brazilian federal government’s legislation portal and state law and municipal law platforms.

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
