# Peer review of "Could Necroleachate Be the Cemetery’s Sewage? A Panorama from Brazilian Legislation"

_ijerph, 2023, doi:10.3390/ijerph20196898_

Round 1

Reviewer 1 Report

The manuscript offers relevant information about the history of cemeteries and the challenges associated with environmental and public health management. However, there are some areas where a critical review can be undertaken to enhance the clarity and fluidity of the text. Here are some observations:

In the abstract, the text effectively introduces the topic of cemetery waste management and its environmental implications in Brazil. To improve, it should provide more specific details about the gaps in legislation, supporting evidence, and potential solutions to address the identified issues.

Introduction:

The text contains some grammatical and spelling errors. For example, "Coemeteriun" should be corrected to "Coemeterium".  A thorough grammar and spelling review is recommended. Some sentences are long and complex, which can hinder comprehension. It is suggested to break down lengthy sentences into shorter ones to improve clarity. It is not clear who the intended audience for this text is. It would be helpful to clearly define the purpose of the text and who it is addressed to (e.g., researchers, public health professionals, policymakers).

Materials and Methods

The methods used and resources employed are clear, making it easy to understand how the study was conducted. The overall paragraph structure is appropriate, with an organized presentation of the steps followed. However, more details about the selection methodology and why specific sources were chosen could be added. Additionally, following formatting and style standards for web link presentations would be beneficial.

Results and Discussion

This section provides detailed information about laws and regulations but lacks a broader context on why these regulations are necessary or what environmental issues they aim to address. The text tends to repeat certain phrases and definitions, such as the definition of solid waste and its classification, multiple times. This redundancy could be reduced to make the text more concise and reader-friendly. The text extensively discusses Brazilian laws and regulations. To broaden its relevance, it could benefit from a brief comparison or reference to how cemetery waste is managed in other countries. This would provide a more global perspective on the topic. The text focuses on definitions and classifications but does not offer recommendations or best practices for effectively managing cemetery waste. Including such guidance could be valuable for readers seeking solutions to this environmental challenge. There is some repetition of information in the text, especially when discussing the composition of necroleachate. Consolidating redundant information could make the text more concise. The text includes numerous citations, adding credibility to the information. However, it could be overwhelming for some readers. Summarizing key findings from these references or providing context for why they are relevant might be beneficial. The text primarily focuses on the situation in Brazil. While this is essential for a Brazilian audience, it might benefit from a brief comparison or reference to how other countries address the issue of necro-leachate, providing a broader perspective. Although the text discusses the problems associated with necro-leachate, it does not offer specific recommendations or solutions to mitigate these issues. Including suggestions for better waste management practices or regulatory changes could enhance the text's practical value.

Final Considerations

The text identifies gaps in legislation but does not offer specific recommendations or solutions to address these issues. Providing suggestions for improving cemetery construction regulations or advocating for changes could enhance the text's practical value.

Reviewer 2 Report

I revised the work and, in my opinion, despite the topic could be interesting, it is now no suitable for publication on this Journal. However, I suggest to the Editor to reconsider your work after strong amendments have been made. English must be carefully revised.

The work is more similar to a discussion paper, so I think that “communication” could be a better type of paper rather that “research article”. Abstract should be more detailed better highlighting the main findings of your work.

In my opinion, the Introduction should be strongly revised. The first part about the story of cemetery,… is completely unnecessary for the scope of the paper related about water pollution and legislation. At the same time, this section lack of explain the novelty of the work and the potential stakeholders of the results.

Moreover, in the text is not well emphasized why the legislation on this topic should be stricter. You write “Necroleachate could be the cemetery's sewage?” in the title. What are typical characteristics of this type of aqueous waste? Main contaminats, concentrations, … in this sense a detailed table ca be helpful.

If the topic of the paper is necroleachate, why discuss also solid waste from cemetery?

Is it a problem common also in other area of the world? It could be helpful to show how it is addressed in other countries (also of the South America) as a comparison.

Moreover, Figures should be named “Figure” and not “graph”. Figure 1 is not clear. What do the percentages refer to?

Conclusions should also report some suggestions about how to tackle the problem. Only report about the need of a stricter and better legislation is not enough. Considering this is the core of the work, for instance, how it can be improved?

Moderate editing of English language required.

Round 2

Reviewer 1 Report

The authors significantly improved the manuscript.

Author Response

We would like to thank you for your efforts with comments and suggestions to improve the manuscript.

Reviewer 2 Report

I thank the authors for their improvements to the manuscript. However, I suggest some minor revisions before possible publication.

- Table 1 is unclear. Please, make the caption clearer. Where are the references used?  Column 1: Why in some case also the city (district) is reported. Please “standardize” the content. Column 2: you add this info to answer my previous comment “Is it a problem common also in other area of the world? It could be helpful to show how it is addressed in other countries (also of the South America) as a comparison.” as reported in the letter of reply. I think that info reported in the column abstract (maybe “notes” is better) seems too general. Please, amend it.

- Section 2: Please, cite the references (also if website) according to the guidelines of the Journal.

- Table 2: Please, change the title “abstract” (e.g., note). In the tables only essential info should be reported. Are the number of the articles an essential info? I think that this column can be removed.

- Table 6: Please, report also in the caption, the references from which the info reported in the table were taken.

Minor editing of English language required.

Author Response

Please see the attachment. We also would like to thank you for your efforts with comments and suggestions to improve the manuscript. 
